# An Analytic Hierarchy Process Method to Evaluate Environmental Impacts of In Situ Oil Shale Mining

**Xiaorong Wang [1,2], Boyue Liu [2,*], Shaolin He [1], Hongying Yuan [2], Dongli Ji [2], Li Qi [2], Yang Song [1] and Wei Xu [1]**

[1]  Beijing Zhonglu Consulting Co., Ltd., PetroChina Planning and Engineering Institute, Beijing 100089, China; wxr747176005@163.com (X.W.); heshaolin0@petrochina.com.cn (S.H.); songyang666@petrochina.com.cn (Y.S.); xuwei_xw@petrochina.com.cn (W.X.)

[2]  School of Environmental and Municipal Engineering, Tianjin Chengjian University, Tianjin 300384, China; yuanhy_00@163.com (H.Y.); yy_cugb@163.com (D.J.); 13821460416@163.com (L.Q.)

*   Correspondence: liuboyue@tcu.edu.cn; Tel.: +86-159-0030-9746

**Abstract:** The great demand for oil shale resource development and the corresponding threats to the environment have resulted in the urgent need to assess the impact of oil shale in situ mining on the environment. In this paper, based on an analysis method developed by the previous literature and the Delphi technique, three secondary indicators and ten tertiary indicators were selected. The weights of the indicators at all levels were subsequently calculated. The results identify environmental capacity, groundwater risk and prevention, and cleaner production as having the largest weights in the indicator system. Following this, the typical three-level indicators with the largest weight and content related to carbon emissions were introduced in detail, and combined with the calculated weight of each indicator, the comprehensive evaluation index method was used to determine the comprehensive evaluation model for the environmental impact of oil shale in situ mining. The comprehensive evaluation model value, $P_A$, of the target layer (the primary indicator) under two different heating methods, combustion heating and electric heating, were then calculated based on the oil shale in situ mining area in Fuyu City, Jilin Province, China. This study introduces carbon-emission-related elements into the three-level indicator evaluation calculation model, which is a more accurate and advanced comprehensive evaluation method.

**Keywords:** analytic hierarchy process; comprehensive model; environmental impact; oil shale in situ mining; index system



## 1. Introduction

Oil shale is a natural solid with a high economic value formed by geological processes in the Earth's crust. It is considered an unconventional oil and gas resource. Oil shale generally exhibits an oil content greater than 3.5% or an organic carbon content greater than 5.0%. Owing to the growing demand for energy, the development and utilization of oil shale is the focus of much international energy development and research [1,2].

There are two principal methods of oil shale mining at present, ground retorting and in situ mining [3]. Ground retorting refers to the process of extracting oil shale underground or in an open pit, crushing and screening it to the required particle size, heating it in an indoor retort, and then producing shale oil or directly using it for combustion and power generation. The development technology of this method is relatively mature, but it produces a large number of polluting gases, sewage, and solid waste. These pollutants will seep into groundwater under the leaching effect of rainwater, resulting in groundwater pollution. In situ mining does not require mining, transportation, or ore processing. Firstly, drilling projects, such as exploration wells, temperature monitoring wells, and production wells, are completed. Then, hydraulic fracturing of the oil shale rock layer using in situ vertical wells is used to generate cracks. The heating device provides a heat source to directly crack the underground oil shale rock layer, which undergoes a pyrolysis reaction

underground, causing the decomposition of kerogen to produce shale oil and shale gas. A large number of microscale pores and fissures are formed along the bedding direction, forming oil and gas communication channels [4,5], achieving the in situ pyrolysis of oil shale and extracting shale oil and gas to the surface through the production well's dry distillation drive. This can develop deep and thick oil shale deposits and has advantages, such as a high recovery rate, good product quality, and less land occupation, and is more environmentally friendly than ground retorting. In situ mining has become a hot spot in the development and utilization of oil shale. In situ mining techniques mainly include combustion heating, electric heating, convection heating, radiation heating, etc. Combustion heating technology is the process of establishing a combustion chamber in a combustion well to heat the oil shale layer, causing oxidation reactions between the dry distilled asphaltene and fixed carbon, thereby providing a heat source for subsequent pyrolysis. Electric heating technology uses an electric heater to heat oil shale layers. Convection heating technology is the process of injecting gas underground to form heat exchange and, thereby, pyrolysis of the oil shale layers. Radiation heating technology utilizes radio frequency heating and supercritical fluids as carriers to achieve oil and gas recovery. After the in situ pyrolysis of oil shale, micro-nanoscale oil and gas permeation channels will be formed inside the rock, improving the permeability of the rock layer. At the same time, it will change the initial stress of the surrounding rock mass. When the stress exceeds the strength of the rock mass itself, the top and bottom rock layers will be destroyed, forming through-cracks and changing the initial flow field of the groundwater [4,6]. Under the action of pore pressure, groundwater can invade the pyrolysis oil shale layers, and toxic and harmful substances, such as inorganic minerals, various organic compounds, and heavy metal elements in the residual oil shale, can migrate and release under the action of water and rock, ultimately affecting the groundwater environment [7,8]. Putra et al. [9] investigated the environmental impact of the in situ pyrolysis of oil shale and reported that the injection of chemical mixtures during the exploitation of oil shale can induce damage to the quality of groundwater close to the exploitation area. Jiang et al. [2] evaluated the impact of oil shale in situ mining on the quality of groundwater and identified a reduction in the groundwater quality in the mining area, with ammonia nitrogen, manganese, and iron as the main overproof ions. Wang et al. [10], in order to explore the impact of solid residue from oil shale pyrolysis on a groundwater environment, conducted ultrapure water–rock interaction experiments. The results show that in situ oil shale mining can lead to continuous pollution of groundwater environments. Hu et al. [11], based on the hydrogeological condition of the oil shale area in Nong'an City, conducted a physical simulation test. It was found the temperature of the surrounding layers continued to be perturbed after the heating of the formation stopped. Zhao et al. [12] conducted pyrolysis and uniaxial compression tests on oil shale samples at different temperatures. The results reveal that as the temperature increased, the elastic modulus and Poisson's ratio of the oil shale exhibited a downward trend, changing the physical and mechanical properties of the oil shale reservoir rock mass and, thus, possibly inducing geological disasters such as land subsidence. In summary, the environmental impact assessment of in situ oil shale extraction is only focused on groundwater, surface water, strata, and other aspects and is lacking a comprehensive environmental assessment. Therefore, in the face of the possible impact of oil shale development and utilization on ecological environments, it is necessary to establish a comprehensive evaluation model for the environmental impact of oil shale in situ mining.

The environmental impact factors of in situ oil shale mining can be considered a multilevel and multiobjective decision-making problem, and scientific methods and evaluation indicators must be used to evaluate the environmental impact caused by in situ mining. Chen et al. [13] evaluated the level of green mine construction by establishing a mixed grey decision model based on the grey analytic hierarchy process and grey clustering method. Shang et al. [14] combined the indicator system, Delphi method, analytic hierarchy process, and fuzzy comprehensive evaluation method to evaluate green phosphate ore. Wang

et al. [15] established evaluation indicators, the comprehensive entropy weight method, and the analytic hierarchy process to conduct a multistandard-level evaluation of green limestone mines. Wang et al. [16] used the AHP to evaluate the regional geological and ecological environmental carrying capacity of Dujiangyan Wenchuan Road. The study showed the spatial distribution of the carrying capacity in this area to be highly concentrated, making it more suitable for planning and construction. Wang et al. combined the fuzzy Delphi method and the fuzzy analytic hierarchy process with geographic information systems to establish a reference framework for the ecological environmental quality of evaluation methods. The experimental results verify the effectiveness and feasibility of this method [17]. Rikhtegar et al. proposed an environmental impact assessment method based on the analysis of network processes and fuzzy simplicity weights to formulate environmental risks for mining projects. The model demonstrates its potential application as a practical problem-solving tool [18]. Aweh et al. used the AHP with multiple criteria to evaluate the weights of selected environmental criteria for the selection of refinery sites. The results reveal the ability of the AHP method to produce reliable and consistent results [19]. The AHP has been reported to be a classic and commonly used method in solving multicriteria decision-making problems [20,21] and is suitable for analyzing the environmental impacts of oil shale in situ mining. In particular, this method breaks down complex problems into ordered hierarchical structures [22], compares the relevant factors of each layer, and evaluates the relative importance and assigned values of each attribute factor. Due to the significant influence of subjective factors on the analytic hierarchy process, this article combines it with the Delphi method and the comprehensive index evaluation method to evaluate the environmental impact of in situ oil shale extraction.

Carbon emissions are a serious impact of human activities on the environment, and the Chinese Ministry of Ecology and Environment has included a carbon emission impact assessment in its environmental impact assessment. The research perspective of carbon emission accounting has been dispersed from an overall perspective to industries such as agriculture, construction, transportation, tourism, and service industries. Determining the impact of changes in material and energy flows on the carbon flow of complex production processes in the iron and steel industry based on the black box model proves to be a difficult task. In particular, it is not possible to derive the carbon emission reduction potential in steel production sites. Based on the requirements of carbon neutrality, energy-intensive industries, such as the iron and steel industry, urgently require accurate carbon accounting [23]. Hydropower is the largest renewable energy source, and its carbon emissions have attracted much attention. However, existing carbon-emission-accounting boundaries are not complete [24]. The carbon emissions from transportation activities in the supply chain are often explained in a relatively simplified manner during the analysis process, with minimal detailed analysis performed. The real impact of transportation on a company's economic and environmental performance is distorted [25]. Therefore, it is necessary to integrate carbon emissions into a comprehensive evaluation model.

At present, research on the environmental assessment of oil shale in situ mining is limited, particularly on the environmental assessment of in situ oil shale mining that incorporates carbon emissions into the evaluation indicators. Therefore, in order to overcome the limitations in the current literature, we integrated the AHP with a literature analysis, the Delphi approach, and the comprehensive evaluation index method to systematically construct a complete comprehensive evaluation model for the environmental impact of oil shale in situ mining. This work provides a decision-making basis for oil shale development project planning and design, as well as environmental protection.

## 2. The Construction Method and Steps of a Comprehensive Evaluation Model for In Situ Oil Shale Mining

A literature analysis was initially employed to determine the indices at all levels, reflecting the impact of oil shale in situ mining on the environment. A hierarchical system at all levels was then established for the preliminarily determined indices by adopting

the Delphi method with expert consultations. The indices and their detailed descriptions were simultaneously improved and modified. Following this, the judgment matrices were determined to calculate the weights of the indices in the criterion and index layer. The literature analysis method was used to construct the evaluation algorithms of the tertiary indices. The comprehensive evaluation index method was subsequently used to interactively evaluate the target layer. Figure 1 presents the specific steps.

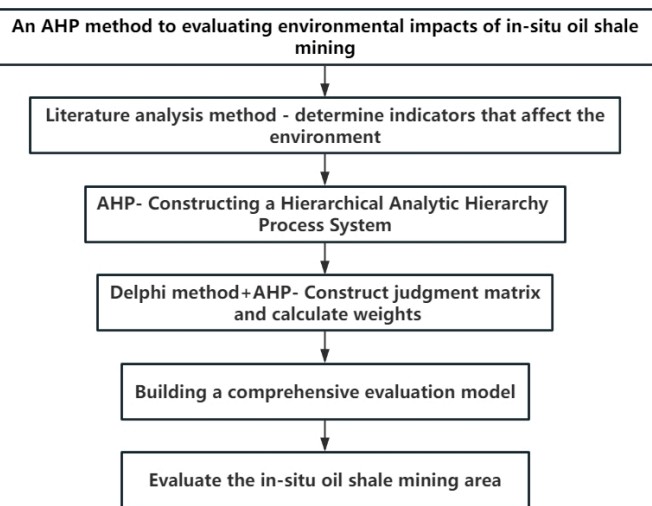

**Figure 1.** Construction of a comprehensive evaluation model.

## 3. Screening of Evaluation Indices

Oil shale in situ mining directly heats the underground oil shale to pyrolyze it underground, and the generated oil and gas are then extracted through production wells. The influence of this process is mainly concentrated underground and has its particularity to the environment. Therefore, according to the inherent characteristics of oil shale in situ mining and the principles of systematicness, operability, and effectiveness, we referred to the existing technical guidelines and other relevant documents for the environmental impact assessments of construction projects in China, such as the "Technical Guidelines for Environmental Impact Assessment: Petrochemical Construction Project", "Technical Guidelines for Environmental Impact Assessment: Onshore Oil and Gas Development", and "Technical Guidelines for Environmental Impact Assessment: Groundwater Environment", to preliminarily determine indicators at all levels. The decomposition and description of key environmental factors were obtained by sorting, analyzing, and summarizing the interactions between the environment and oil shale in situ mining. An expert consultation meeting was then held to supplement and adjust the proposed factors and remove those deemed as unnecessary. The environmental comprehensive evaluation model for oil shale in situ mining included three secondary indices and ten tertiary indices that were determined at a second expert consultation meeting.

The second-level indicator of environmental site selection refers to the need to consider the hydrological environment, geological environment, environmental carrying capacity, and special ecological protection areas of the mining area before in situ oil shale mining. By comprehensively considering the environmental conditions of the mining area and the advantages and disadvantages of the mining area, site selection is carried out. The evaluation is mainly carried out through four three-level indicators: hydrogeology, engineering geology, ecological sensitivity, and environmental capacity.

The second-level indicator of environmental risk refers to the degree and possibility of harm to the environment caused by sudden accidents during oil shale in situ mining activities. The evaluation is mainly conducted through three three-level indicators: groundwater risk and prevention, environmental geological risk and prevention, and other environmental risks and prevention.

The secondary indicator of environmental governance refers to the use of governance measures to control the entire mining process (construction, production, and retirement periods) of oil shale in situ mining projects, reducing the impact of the project on the ecological environment. The evaluation is mainly conducted through three three-level indicators: clean production, pollution control, and process control. Figure 2 depicts the specific hierarchy of the system.

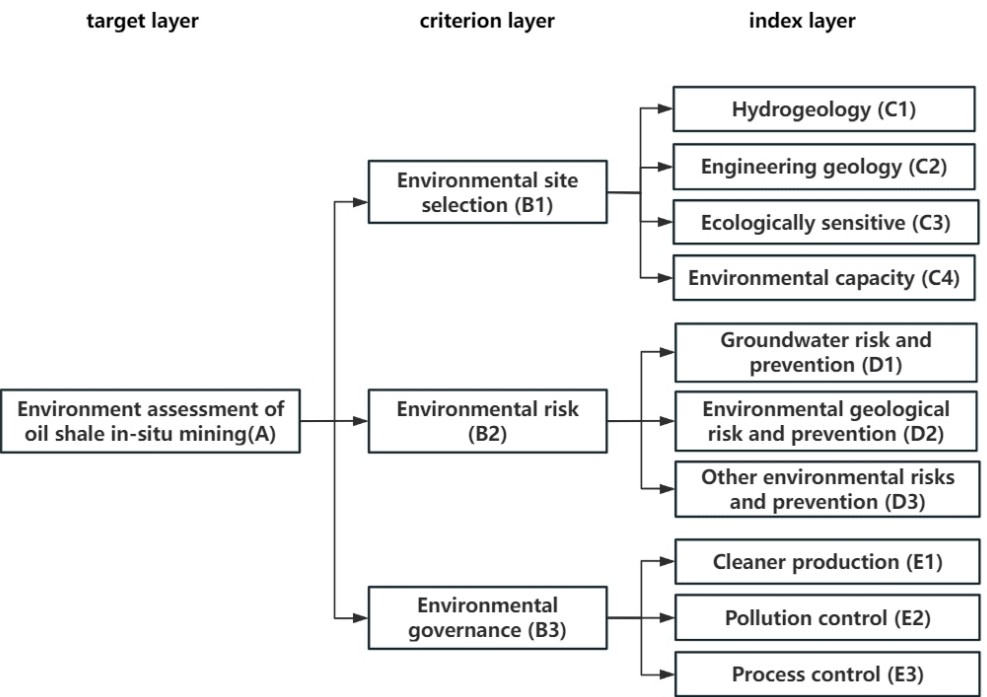

**Figure 2.** Hierarchical index system of environmental assessment of oil shale in situ exploitation.

The target layer (primary indices) contains the environmental assessment of oil shale in situ mining (A); the criterion layer (secondary indices) contains environmental site selection (B1), environmental risk (B2), and environmental governance (B3); and the index layer (tertiary indices) contains hydrogeology (C1), engineering geology (C2), ecological sensitivity (C3), environmental capacity (C4), groundwater risk and prevention (D1), environmental geological risk and prevention (D2), other environmental risks and prevention (D3), cleaner production (E1), pollution control (E2), and process control (E3). The fourth-level indicator is represented by "m".

## 4. Calculation of Index Weights Using the Analytical Hierarchy Process

The environmental impacts of oil shale exploitation are related to geology, oil and gas exploitation, hydrogeology, groundwater protection, wastewater treatment, air pollution control, solid waste disposal, environmental planning, environmental impact assessments, and other disciplines. In order to ensure that the determination of the indices and weights was more comprehensive, scientific, and reasonable while also considering economic feasibility, 15 scholars and experts with senior professional titles were selected from universities and enterprises. All experts had worked with environmental impact assessments for over 10 years. According to the Delphi method, the survey form was first distributed to each expert (refer to Supplementary Material S1). A total of 15 questionnaires were returned, and the opinions of various experts were summarized (refer to Supplementary Material S2) and listed in a table. The obtained data table was converted into a three-scale judgment matrix.

### 4.1. Construction of Judgment Matrices

This study performed a preliminary analysis on the relationship between the indices in the environment evaluation system of oil shale in situ mining and constructed a pairwise

comparison judgment matrix for the factors at the same stratum. In order to quantitatively display the importance of each element in the matrix, the three-scale method was introduced to construct the judgment matrix. First, experts judged the importance of two indices according to the three-scale method until the consistency of the judgment matrix was achieved. The factor weights were further determined by the corresponding transformed 1–9 scale matrix. The maximum eigenvalue of the judgment matrix was obtained, and the consistency was tested in order to increase the feasibility of the index weights [26]. Finally, the weight of each index was modified. The modified weights can effectively integrate the opinions of experts and unify the opinions of experts [27].

*4.2. Determination of Index Weights*

After converting the three-scale method into the 1–9 scale judgment matrix, the index weights were calculated. Table 1 reports the results.

**Table 1.** Weights of the comprehensive evaluation model for the environmental impact of oil shale in situ mining.

| Target Layer | Criterion Layer (Secondary Indices) | Secondary Index Weight | Index Layer (Tertiary Indices) | Tertiary Index Weight |
|---|---|---|---|---|
| The comprehensive evaluation model for the environmental impact of oil shale in situ mining | Environmental site selection B1 | 0.464 | Hydrogeology C1 | 0.197 |
| | | | Engineering geology C2 | 0.198 |
| | | | Ecologically sensitive C3 | 0.253 |
| | | | Environmental capacity C4 | 0.352 |
| | Environmental risk B2 | 0.289 | Groundwater risk and prevention D1 | 0.515 |
| | | | Environmental geological risk and prevention D2 | 0.285 |
| | | | Other environmental risks and prevention D3 | 0.200 |
| | Environmental governance B3 | 0.247 | Cleaner production E1 | 0.385 |
| | | | Pollution control E2 | 0.257 |
| | | | Process control E3 | 0.358 |

As shown in Table 1, the weight of the environmental site selection (B1) in the criterion layer was determined as 0.464, while the environmental risk and environmental governance weights were 0.289 and 0.247, respectively. Therefore, in the criterion layer, the environmental site selection (B1) was the most important factor, and environmental governance (B3) was the least. The tertiary indices with the largest weight in the environmental site selection (B1), environmental risk (B2), and environmental governance (B3) were environmental capacity (C4, 0.352), groundwater risk and prevention (D1, 0.515), and cleaner production (E1, 0.385), respectively.

**5. Construction of Typical Indices and Comprehensive Evaluation Model**

We combined the relevant technical guidelines with the index weights in Table 1 to comprehensively evaluate the target layer by constructing a comprehensive evaluation model using the comprehensive evaluation index method. Through the construction of the comprehensive evaluation model, the limitations of the more statutory index of the AHP were overcome, and the quantitative evaluation of the environmental impact of oil shale in situ mining was realized [28].

*5.1. Construction of an Evaluation Model of Typical Tertiary Indices*

Environmental capacity (C4), groundwater risk and prevention (D1), and clean production (E1) account for a large proportion of the third-level index, indicating that these indicators have a large impact on the standard layer. Moreover, greenhouse-gas-related content is added to the four third-level indicators, namely, environmental capacity (C4),

clean production (E1), pollution control (E2), and process control (E3). Therefore, we constructed these typical three-level indicator comprehensive evaluation models and applied the comprehensive evaluation index method to evaluate the target layer.

### 5.1.1. Environmental Capacity (C4)

The evaluation of environmental capacity was performed through the absolute capacity, which is defined as the maximum load of a pollutant that can be accommodated in the oil shale mining area, assuming that the accumulated concentration of pollutants does not exceed the maximum allowable value specified in the environmental standard [29]. This can be divided into surface water environmental capacity (m1), atmospheric environmental capacity (m2), and carbon fixation (m3) based on the environmental factors.

(1)　Weights of environmental capacity sub-indices

Refer to Supplementary Section: 5.1.1 Environmental Capacity, (1) Weights of environmental capacity sub-indices;

(2)　Assessment and evaluation method

Refer to Supplementary Section: 5.1.1 Environmental Capacity, (2) Assessment and evaluation method;

(3)　Environmental capacity assessment model ($P_{C4}$)

Table S1 reveals the sub-index assessment values of environmental capacity to be surface water environmental capacity (m1), atmospheric environmental capacity (m2), and carbon fixation (m3). Thus, the evaluation model of environmental capacity is described as:

$$P_{C4} = 0.3 \times m1 + 0.5 \times m2 + 0.2 \times m3 \tag{1}$$

### 5.1.2. Groundwater Risk and Prevention (D1)

Groundwater risk and prevention were evaluated through two sub-indices, namely, groundwater risk potential (m1) and groundwater risk prevention (m2). By setting the evaluation interval level, the evaluation values were divided into two sub-indices, and finally, the evaluation values of groundwater risk and prevention were obtained as the weighted average.

(1)　Sub-index weights of groundwater risk and prevention

Refer to Supplementary Section: 5.1.2 Groundwater Risk and Prevention, (1) Sub-index weights of groundwater risk and prevention;

(2)　Assessment and evaluation method

Refer to Supplementary Section: 5.1.2 Groundwater Risk and Prevention, (2) Assessment and evaluation method;

(3)　Groundwater risk and prevention evaluation model ($P_{D1}$)

Table S3 reveals the assessment values of groundwater risk potential (m1) and groundwater risk prevention (m2). Thus, the evaluation model of groundwater risk and prevention is described as:

$$P_{D1} = 0.7 \times m1 + 0.3 \times m2 \tag{2}$$

### 5.1.3. Cleaner Production (E1)

The evaluation of cleaner production was determined according to the impact of the mining indicators on the real benefits and level of cleaner production of oil shale exploration, as well as the development enterprises and the difficulty of implementation. The following six indices were used for the evaluation: comprehensive energy consumption per unit product (m1); characteristics of production equipment (m2); waste oil recovery rate (m3); wastewater compliance rate (m4); production management system (m5); and carbon emissions (m6).

(1)  Weight of cleaner production sub-indices

Supplementary Section: 5.1.3 Cleaner production, (1) Weight of cleaner production sub-indices, Table S7 Weights of cleaner production sub-indices;

(2)  Assessment and evaluation method

Accounting of carbon emissions

We adopted the life cycle concept for the carbon emission accounting of the oil shale in situ mining area [30]. The carbon emission accounting model is established through five production links, including drilling, hydraulic fracturing, steam injection heating, oil and gas collection, separation and emission, and transportation to the refinery, as well as five carbon emission sources, including fuel, escape, oil and gas collection, electricity, and water resources. This allows for the maximization of emission accounting during the entire production cycle of oil shale in situ mining and improves the accounting accuracy. Figure 3 presents the technical process.

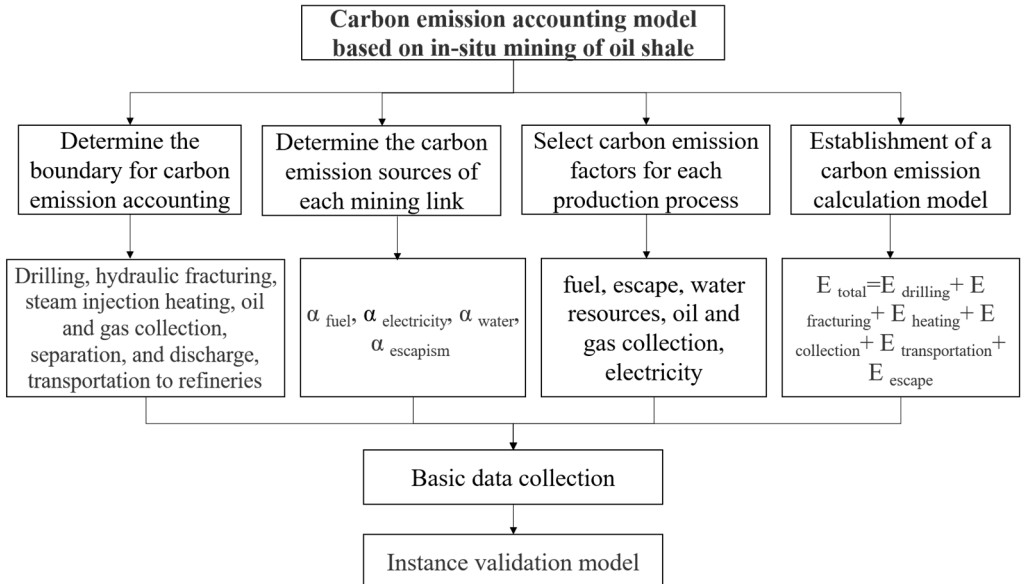

**Figure 3.** Technical flowchart of the carbon emission accounting method for the oil shale in situ mining area.

Prior to carbon emission accounting, the accounting boundary was determined as follows: drilling in the in situ exploitation of oil shale; hydraulic fracturing; steam injection heating; oil and gas collection; treatment and emission; and transportation to the refinery. Five types of carbon emission sources were then identified based on the fuel combustion, escape, process emissions, power grid, and water resources in each exploitation process. The carbon emission factors corresponding to each carbon emission source were then selected.

The drilling link in the oil shale in situ mining process includes the consumption of diesel oil by motor equipment and the energy consumption of electric-powered large equipment and water resources. Hydraulic fracturing includes the fuel consumption of diesel oil and raw coal, the power consumption of high-pressure pumps, and water resource consumption. The steam injection heating link includes the energy consumption of diesel oil and raw coal, the power consumption of water pumps and other equipment, the consumption of water resources, and the greenhouse gases generated from the decomposition of organic matter, carbonate, etc., during the heating process. The links of oil and gas collection, treatment, and emissions include the consumption of diesel, electricity, and water resources in the processes of oil and gas condensation, oil and water separation, gas collection, and the emissions of collected gas. The links from transportation to the refinery include the consumption of diesel, electricity, and water resources in the process of vehicle and pipeline transportation. All of the aforementioned factors are within the

scope of carbon emissions accounting for the ecological environment of the oil shale in situ mining area.

The carbon emission factors of raw coal consumption ($\alpha$raw coal), diesel consumption ($\alpha$diesel), power grid consumption ($\alpha$electricity), and water resource consumption ($\alpha$water), as well as the fugitive carbon emission factors ($\alpha_{CH4}$ escape and $\alpha_{CO2}$ escape), were determined and corrected with reference to the relevant data and are shown in Table S9.

Refer to attached document: 5.1.3 Cleaner production, (2) Assessment and evaluation method, Table S8 Carbon emission factors corresponding to different energy sources.

The carbon emissions produced by raw coal consumption ($E_{raw\ coal}$), diesel fuel ($E_{diesel\ fuel}$), grid power consumption ($E_{electric\ power}$), and water resource consumption ($E_{water}$) were calculated as follows:

$$E_i = M_i \times \alpha_i \tag{3}$$

where i is raw coal/diesel oil/electric power/water resources; $M_{raw\ coal}$ is the consumption of raw coal; $M_{diesel\ oil}$ is the consumption of diesel oil; $M_{electricity}$ is the consumption of electric power in the power grid; and $M_{water}$ is the consumption of water resources.

The carbon emissions from drilling, hydraulic fracturing, steam injection heating, oil and gas collection, treatment and emissions, and transportation to the refinery in the mining area were calculated as follows:

$$E_{well\ drilling/transport} = E_{diesel\ oil} + E_{electric\ power} + E_{water} \tag{4}$$

$$E_{fracture/heating} = E_{raw\ coal} + E_{diesel\ oil} + E_{electric\ power} + E_{water} \tag{5}$$

$$E_{collect} = E_{emission} + E_{electric\ power} + E_{water} \tag{6}$$

Following this, based on the production of oil shale in the mining area (M oil shale), we determined the total amount of escape during hydraulic fracturing, steam injection heating, and oil and gas collection ($E_{escape}$) as follows:

$$E_{escape} = E_{well\ drilling} + E_{fracture} + E_{heating} + E_{collect} + E_{transport} \tag{7}$$

where $GWP_{CH4}$ is the global warming potential value of $CH_4$; $\rho_{CH4}$ is the density of $CH_4$; and $\rho_{CO2}$ is the density of carbon dioxide.

Thus, the total carbon emissions were determined as:

$$E_{total} = GWP_{CH4} \times M_{oil\ shale} \times \alpha_{CH4\ escape} \times \rho_{CH4} + M_{oil\ shale} \times \alpha_{CO2\ escape} \times \rho_{CO2} \tag{8}$$

Refer to attached document: 5.1.3 Cleaner production, (2) Assessment and evaluation method, Table S9 Classification of the assessment value of the cleaner production sub-indices;

(3) Cleaner production evaluation model ($P_{E1}$)

Table S7 reveals the sub-index assessment values of cleaner production as the comprehensive energy consumption per unit product (m1), production equipment characteristics (m2), waste oil recovery rate (m3), wastewater discharge rate up to standard (m4), production management system (m5), and carbon emissions (m6). Thus, the evaluation model of cleaner production was based on the following:

$$P_{E1} = 0.21 \times m1 + 0.21 \times m2 + 0.16 \times m3 + 0.16 \times m4 + 0.1 \times m5 + 0.16 \times m6 \tag{9}$$

5.1.4. Pollution Control (E2)

The assessment of pollution control was based on treatment technology that eliminates or reduces the impact of pollutants on the environment, with the four sub-indicators of

water pollution control (m1), air pollution control (m2), solid waste disposal (m3), and carbon reduction performance (m4).

(1)    Sub-index weights of the pollution index

Supplementary Section: 5.1.4 Pollution control, (1) Sub-index weights of the pollution index;

(2)    Assessment and evaluation methods

① Water pollution control (m1)

The treatment of water pollution denotes the production of sewage (waste) water treatment plants or facilities to treat the water pollutants generated in the oil shale mining process and is mainly reflected in the rate of reaching the effluent quality standard. The assessment value of the effluent pollution treatment was obtained according to the rate of reaching the effluent quality standard;

② Air pollution control (m2)

The atmospheric pollutants generated during oil shale exploitation are generally treated through the implementation of an air pollution control project, which is mainly reflected in the rate of reaching the air pollutant discharge standard. The assessment value of air pollution control was, therefore, obtained according to the rate of reaching the air pollutant discharge standard;

③ Solid waste disposal (m3)

Solid waste disposal is safe if certain methods are adopted and is mainly reflected in the rate of reaching the solid waste standard. The assessment value of solid waste disposal is, thus, obtained according to the rate of reaching the solid waste standard;

④ Carbon reduction performance (m4)

The carbon emission reduction approaches of oil shale in situ mining generally focus on energy conservation and consumption reduction, the development of low-carbon energy, the capture and utilization of terminal $CO_2$, increasing carbon sinks, waste utilization, etc., all of which are described in the following:

Energy saving and consumption reduction: Oil shale in situ mining areas consume a large amount of diesel during transportation, resulting in a large amount of carbon emissions. Thus, appropriate measures must be taken to reduce diesel consumption, such as maintaining roads, increasing road repair efforts, shortening transportation distance, strengthening the management of vehicles, the rational use of vehicles, reducing idle fuel consumption, etc. Moreover, controlling high-energy consumption equipment in each mining link, improving the equipment process level, and selecting advanced equipment to reduce energy consumption can also facilitate diesel consumption reductions.

Development of low-carbon energy: Oil shale in situ mining areas can use natural gas, a relatively clean energy, to replace coal and fuel oil. This is an important direction for the development of carbon energy and is mainly reflected in the reduction of carbon dioxide emissions in the power industry by replacing coal with natural gas, wind energy, solar energy, etc. In the process of replacing coal for power generation, natural gas can reduce carbon dioxide emissions by 60%. This is the principal cause of the reduction in carbon dioxide emissions in the United States in recent years [31]. The proportion of wind and solar power generation is projected to gradually increase in the future.

Capture and utilization of terminal $CO_2$: Numerous links in the process of oil shale in situ mining, including site development, drilling, hydraulic fracturing, steam injection heating, oil and gas collection, treatment and emissions, transportation to the refinery, wastewater treatment, and domestic waste, produce a large amount of $CO_2$, which will accelerate greenhouse gases when discharged into the atmosphere. $CO_2$ fracturing technology is not only applicable to low-permeability complex rock formations, but when $CO_2$ is injected into the formation for fracturing, it can also utilize the waste $CO_2$ generated in the mining process and reduce the emissions of greenhouse gases into the atmosphere, thus indirectly reducing carbon emissions [32,33]. $CO_2$ flooding technology buries a large amount of $CO_2$ underground and improves oil and gas recovery.

Increasing carbon sinks: In situ oil shale mining areas have changed the land use types of sites during development, damaged land cover, greatly reduced the carbon fixation capacity of the land, and increased carbon emissions. The restoration of the ecological environment (e.g., soil and vegetation restoration) in the mining area can effectively improve carbon sinks. Based on this, the amount of soil carbon and vegetation carbon sequestration in mining areas can be effectively improved by ecological restoration and vegetation and soil reconstruction. In addition, some land is left unused in the process of site development. Clean energy (e.g., solar and wind energy) can be developed on idle land, thus reducing the use of fossil fuels and carbon emissions [34].

Waste utilization: The solid waste and domestic garbage generated from the in situ mining of oil shale are usually disposed of by landfilling or incineration, which generates a large amount of greenhouse gases, such as $CO_2$ and $CH_4$. Waste utilization can effectively achieve carbon emission reductions. For example, ash can be used for cement construction materials. Drilling, hydraulic fracturing, steam injection heating, and other links in the process of oil shale in situ mining can produce a large amount of wastewater. Reusing this wastewater for fracturing in the injection well can reduce the consumption of water resources and reduce carbon emissions;

⑤ Assessment value of the pollution control sub-indices

Refer to Supplementary Section: 5.1.4 Pollution control, (2) Assessment and evaluation methods

(3)　Pollution control evaluation model $P_{E2}$

Table S10 reveals the sub-index values of pollution control as water pollution control (m1), air pollution control (m2), solid waste disposal (m3), and carbon reduction performance (m4). The assessment model of pollution control is, thus, described as:

$$P_{E2} = 0.45 \times m1 + 0.25 \times m2 + 0.2 \times m3 + 0.1 \times m4 \tag{10}$$

5.1.5. Process Control (E3)

The process control evaluation was based on the management technology of the whole project during and after the mining process, mainly through three sub-indicators: monitoring management (m1), equipment management (m2), and decommissioning control (m3).

(1)　Process control sub-index weights

Refer to Supplementary Section: 5.1.5 Process control, (1) Process control sub-index weights;

(2)　Assessment and evaluation methods

Carbon emission monitoring and control

Internationally recognized greenhouse gas quantification methods can be divided into the material balance algorithm, emission factor approach, and measurement method. Among them, the material balance algorithm and emission factor approach are calculation methods, while the measurement method involves the monitoring of emissions using online monitoring instruments.

The accuracy of carbon emission management depends on the carbon emission monitoring system. Compared with mature environmental monitoring approaches, the carbon emission monitoring system is more complex. In particular, it focuses on the accounting of carbon dioxide, including the direct and indirect emissions of carbon dioxide. It can monitor energy consumption in real time and convert it into real-time carbon emissions according to local conditions. Moreover, the carbon emission monitoring system has a more complete response plan and additional measures for carbon emissions exceeding the standard.

At present, there are two measurement methods for online carbon emission monitoring systems, namely, in situ measurements and sampling with pretreatment. The measurement method of sampling and pretreatment can overcome the limitations of the in situ measurement method. The sampling probe samples the flue gas, which travels to the analysis

cabinet through the sampling and heat-tracing pipelines. After mixing, pretreatment, and measurements, the $CO_2$ concentration is obtained. The $CO_2$ concentration, flow, pressure, temperature, and other parameters are transferred to the carbon-emission monitoring-data management and analysis expert system for comprehensive processing, and the $CO_2$ emission rate and emissions of the enterprise are calculated. Notably, novel cost-effective, intelligent approaches have emerged to tackle issues linked to the monitoring of different parameters [35,36].

As well as the strict monitoring of carbon emissions, emergency plans should be developed to deal with the risk of carbon emissions exceeding the standard, for example, rapid-response strategies to determine the causes and main links for cases exceeding the standard. Further approaches for important links with high carbon emissions include the timely reduction of energy consumption, suspending production activities with high greenhouse gas emissions, organizing expert meetings, optimizing technology and improving the management system, equipment maintenance and upgrading, strengthening the control of the production process, eliminating the risk of carbon emissions exceeding the standard caused by process-equipment problems or illegal operations, and ensuring that carbon emissions meet the original design requirements.

Supplementary Section: 5.1.5 Process control, (2) Assessment and evaluation methods, Table S13 Grading of process control assessment values;

(3)     Process control evaluation model $P_{E3}$

Table S12 reveals monitoring management (m1), equipment management (m2), and retirement management and control (m3) to be the sub-indicators of process control. Thus, the evaluation model of process management and control is defined as:

$$P_{E3} = 0.6 \times m1 + 0.1 \times m2 + 0.3 \times m3 \tag{11}$$

*5.2. Construction of a Comprehensive Evaluation Model*

The comprehensive evaluation model for the environmental impact of oil shale in situ mining is expressed as:

$$A = \sum_{i=1} W_i \sum_{j=1}^{m} w_{ij} \times P_{ij} \tag{12}$$

where $W_i$ is the weight of the secondary index i relative to the target layer; i is C, D, and E; m is the total number of tertiary indices in the secondary index i; $W_{ij}$ is the weight of the tertiary index j selected from the secondary index i relative to the second-level index i; $P_{ij}$ is the evaluation model value of the tertiary index j selected from the second-level index i.

Based on the weights of the secondary and tertiary indices calculated from the evaluation models of each tertiary index, the comprehensive evaluation model $P_A$ of the system can be obtained as follows:

$$\begin{aligned} PA = {} & 0.464 \times (0.197PC1 + 0.198PC2 + 0.253PC3 + 0.352PC4) + 0.289 \times (0.515PD1 \\ & + 0.285PD2 + 0.2PD3) + 0.247 \times (0.385PE1 + 0.257PE2 + 0.35PE3) \end{aligned} \tag{13}$$

We can, thus, evaluate the impact of oil shale in situ mining projects on the environment using the comprehensive evaluation model constructed in this study. This allows us to determine whether the environmental impact is feasible for oil shale mining projects.

## 6. Calculation of Comprehensive Evaluation Model

This paper selected the Changchunling oil shale mining area in Fuyu City, Jilin Province, China, to verify the comprehensive evaluation model of the environment for in situ mining and compares the evaluation results of the mining area under two different heating methods: combustion heating and electric heating. Grassland, industrial land, storage land, residential land, rural roads, and vacant land cover an area of 85,836, 77,190, 500, 300, 28,950, and 200 $m^2$, respectively, in the study area, with a total area of 192,976 $m^2$. We calculated the comprehensive evaluation model based on the real situation of the mining area, referring to the relevant standards. The following focuses on the calculation process of

typical tertiary indicators under the combustion heating mode; for the calculation process of the other indicators, please refer to the attached document: Section 6, Calculation of comprehensive evaluation model. The calculation of the electric heating mode may be obtained similarly.

*6.1. Environmental Capacity (C4)*

(1) Surface water environmental capacity (m1): We refer to the "Surface Water Environmental Quality Standard" (GB 3838-2002) [37] and relevant information. The assessment score for m1 is determined as 70;

(2) Atmospheric environmental capacity (m2): We refer to the "Air Environmental Quality Standard" (GB 3095-2012) [38] and relevant information. The assessment score for m2 is calculated as 65;

(3) Carbon sequestration (m3): The sum of carbon sequestration in forest and arable land is the total amount of carbon sequestration. The assessment score for m3 is determined as 70.

The final environmental capacity assessment model is derived as $P_{C4} = 0.3 \times m1 + 0.5 \times m2 + 0.2 \times m3 = 67.5$.

*6.2. Groundwater Risk and Prevention (D1)*

(1) Groundwater risk potential (m1): According to the "Groundwater Quality Standard" (GB/T14848-93) [39], the groundwater functional area in the mining area is determined as a Class III water body functional area, with the assessment score for m1 equal to 80;

(2) Groundwater risk prevention (m2): During the pyrolysis process, the number of reservoir fractures increases. During lifting and operation, attention should be paid to optimizing the design of the inflow fluid to avoid any reservoir pollution caused by the filtration and loss of the inflow fluid. Furthermore, after the completion of the mining process, each wellbore should be filled to prevent groundwater from entering the wellbore and being polluted or polluted groundwater from replenishing deeper groundwater, thus contaminating deeper groundwater. The assessment score for m2 is determined as 70.

The final groundwater risk and prevention evaluation model is derived as $P_{D1} = 0.7 \times m1 + 0.3 \times m2 = 77$.

*6.3. Clean Production (E1)*

(1) Comprehensive energy consumption per unit product (m1): We used Comprehensive energy consumption per unit product = Comprehensive energy consumption per product output. The assessment score for m1 is determined as 65;

(2) Production equipment characteristics (m2): The mining area adopts underground in situ cracking technology, which has strong heating and combustion effects, a high resource utilization rate, and an advanced production process and equipment, and meets the expected quality requirements. Each platform is well-equipped with a set of online monitoring devices for oxygen content. A two-phase separator is used for oil and gas separation to realize the fully enclosed gathering and transmission process in the station. The dehydration index of oil extracted from shale oil in a multipurpose station should reach the depth of purified oil. A high-efficiency electromagnetic coalescence electric dehydrator is the preferred dehydration equipment for multipurpose stations due to its high automation. The assessment score for m2 is determined as 90;

(3) Sewage oil recovery rate (m3): The sewage oil recovery rate in the in situ mining area reaches 80%, and the assessment score for m3 is 80;

(4) Wastewater compliance rate (m4): The wastewater generated in the mining area is sent to the sewage treatment station of Jilin Songyuan Petrochemical Co. Ltd., for treatment, and the effluent quality meets the "Comprehensive Wastewater Discharge Standard" (GB8978-1996) [40]. It is then discharged into the Jiangbei Sewage Treatment Plant of

Songyuan City for further treatment and discharge after reaching the standard. The assessment score for m4 is 80;

(5) Production management system (m5): In determining the production process and equipment selection process, the mining area strictly follows the rational utilization of resources and rational use of energy and maximizes energy conservation. The assessment score for m5 is 80;

(6) Carbon emissions (m6): Each ton of oil extracted from shale oil produced in the mining area produces 4.9 tons of $CO_2$, and the assessment score for m6 is 61.

The final clean production evaluation model is derived as $P_{E1}$ = 0.21 × m1 + 0.21 × m2 + 0.16 × m3 + 0.16 × m4 + 0.1 × m5 + 0.16 × m6 = 71.

### 6.4. Pollution Control (E2)

(1) Water pollution control (m1): The mining area adopts a separate drainage system, and the extracted liquid separation wastewater, boiler discharge wastewater, desulfurization wastewater, and waste fracturing fluid backflow liquid are sent to the sewage treatment station of Jilin Songyuan Petrochemical Co., Ltd. for treatment. The effluent quality meets the "Comprehensive Wastewater Discharge Standard" (GB8978-1996) [40]. It is then discharged into the Jiangbei Sewage Treatment Plant of Songyuan City through the sewage pipe network for further treatment and discharge after reaching the standard. The assessment score for m1 is 80;

(2) Air pollution control (m2): The waste gas from mining areas includes boiler flue gas, torch gas, and unorganized waste gas. The flue gas emission concentration of the heating boiler meets the emission concentration limit requirements of the newly built gas boiler in the Emission Standard of Air Pollutants for Boilers (GB13271-2014) [41]. Shale gas is used as fuel for boiler combustion during the heating period, and the separated shale gas is burned using a flare incineration system during the non-heating period. The hydrocarbon gases lost through unorganized volatilization must meet the concentration standards for unorganized monitoring in the "Comprehensive Emission Standards for Air Pollutants" (GB16297-1996) [42]. The assessment score for m2 is 75;

(3) Solid waste disposal (m3): Dangerous substances, such as oil sludge and waste anti-seepage cloth, generated in mining areas are regularly sent to units with hazardous waste treatment qualifications for treatment. The storage, transfer, and transportation of hazardous waste must strictly comply with the requirements of the "Pollution Control Standards for Hazardous Waste Storage" (GB18597-2001) [43] and the "Management Measures for Hazardous Waste Transfer Manifests". Production waste is collected in a centralized manner and regularly sent to nearby urban landfill sites for treatment. The assessment score for m3 is 70;

(4) Carbon reduction performance (m4): According to the carbon emissions generated by the production of each ton of oil extracted from shale oil in the mining area, carbon emission reduction approaches typically focus on energy conservation and consumption reduction, the development of low-carbon energy, the capture and utilization of terminal $CO_2$, increasing carbon sinks, waste utilization, etc. The assessment score for m4 is determined as 80.

The final pollution control evaluation model is derived as $P_{E2}$ = 0.45 × m1 + 0.25 × m2 + 0.2 × m3 + 0.1 × m4 = 76.8

### 6.5. Process Control (E3)

(1) Monitoring management (m1): Monitoring wells deployed in the mining area are mainly used for dynamic monitoring production performance, such as combustion front monitoring in the well cluster, oil shale temperature distribution monitoring, and groundwater movement and water quality monitoring in the mining area, to provide a basis for production adjustment and environmental protection. For carbon emission monitoring and control, energy consumption can be monitored in real time and converted into real-time carbon emissions based on local conditions. The mining

area has a relatively complete response plan and measures for carbon emissions that exceed standards. The assessment score for m1 is 75;

(2) Equipment management (m2): The production process and equipment are advanced, and regular maintenance and repair are carried out. The assessment score for m4 is 75;

(3) Retirement control (m3): Measures include the dismantling and cleaning of retired equipment, the cleaning and decontamination of process pipelines, and the transfer to professional companies to organize retirement. The assessment score for m3 is determined as 65.

The final process control evaluation model is derived as $P_{E3} = 0.6 \times m1 + 0.1 \times m2 + 0.3 \times m3 = 72$.

The evaluation results of various indicators under the combustion heating method are shown in Table 2. The highest score for the hydrogeological assessment is 88, and the lowest score for the ecological sensitivity assessment is 48.1. It is necessary to improve the third-level indicator of ecological sensitivity. The final comprehensive evaluation model obtained a PA of 72.6. Similarly, the PA value under the electric heating method is 70.5. (The evaluation results are shown in Table 3.) The results indicate that the environmental impact of in situ mining under the combustion heating method in the region is smaller than that under the electric heating method. In China, there are few pilot projects for in situ oil shale extraction, and the corresponding data for real extraction are also difficult to obtain. The comprehensive evaluation model in this article mainly focuses on the evaluation of different heating methods for the oil shale pilot demonstration base in Fuyu City, Jilin Province. This is only a preliminary study, and the comprehensive evaluation model still needs to be improved, but it can be used as a reference for subsequent research. In the future, with the development of the oil shale industry, the number of in situ mining areas will gradually increase. We will further improve the model, verify its applicability, and compare and evaluate multiple mining areas.

**Table 2.** Model evaluation values of indicators at all levels under combustion heating mode.

| Level 3 Indicator | Level-4 Indicator Assessment Value | | | | | | Level-3 Indicator Assessment Value | $P_A$ |
|---|---|---|---|---|---|---|---|---|
| | m1 | m2 | m3 | m4 | m5 | m6 | | |
| Hydrogeology (C1) | 100 | 70 | — | — | — | — | 88 | |
| Engineering geology (C2) | 68 | 81.4 | 72.2 | 88.6 | — | — | 77 | |
| Ecological sensitivity (C3) | 59.1 | 44.5 | 0 | 26.2 | — | — | 48.1 | |
| Environmental capacity (C4) | 70 | 65 | 70 | — | — | — | 67.5 | |
| Groundwater risk and prevention (D1) | 80 | 70 | — | — | — | — | 77 | 72.6 |
| Environmental geological risks and prevention (D2) | 80 | 90 | 80 | — | — | — | 83 | |
| Other environmental risks and prevention (D3) | 75 | 75 | 80 | 85 | — | — | 78 | |
| Clean production (E1) | 65 | 90 | 80 | 80 | 80 | 61 | 71 | |
| Pollution control (E2) | 80 | 75 | 70 | 80 | — | — | 76.8 | |
| Process control (E3) | 75 | 75 | 65 | — | — | — | 72 | |

**Table 3.** Model evaluation values of indicators at all levels under electric heating mode.

| Level 3 Indicator | Level-4 Indicator Assessment Value | | | | | | Level-3 Indicator Assessment Value | $P_A$ |
|---|---|---|---|---|---|---|---|---|
| | m1 | m2 | m3 | m4 | m5 | m6 | | |
| Hydrogeology (C1) | 100 | 70 | — | — | — | — | 88 | |
| Engineering geology (C2) | 68 | 81.4 | 72.2 | 88.6 | — | — | 77 | |
| Ecological sensitivity (C3) | 59.1 | 44.5 | 0 | 26.2 | — | — | 48.1 | |
| Environmental capacity (C4) | 70 | 65 | 70 | — | — | — | 67.5 | |
| Groundwater risk and prevention (D1) | 80 | 80 | — | — | — | — | 80 | 70.5 |
| Environmental geological risks and prevention (D2) | 80 | 90 | 80 | — | — | — | 83 | |
| Other environmental risks and prevention (D3) | 75 | 75 | 80 | 90 | — | — | 79 | |
| Clean production (E1) | 40 | 50 | 70 | 70 | 60 | 30 | 52.1 | |
| Pollution control (E2) | 80 | 65 | 70 | 70 | — | — | 73.3 | |
| Process control (E3) | 70 | 75 | 55 | — | — | — | 66 | |

In order to achieve the sustainable development of oil shale in situ mining areas, reduce damage to the environment of mining areas [31], and protect important sources of surface water and groundwater, it is necessary to plan mining conditions and processes reasonably and improve industrial practices in the oil shale industry [32], and all levels of managerial departments should also attach importance to the planning and governance of environmental quality in mining areas, promoting the continuous improvement of environmental quality.

## 7. Conclusions

(1) The comprehensive evaluation model for the environmental impact of oil shale in situ mining was established based on the characteristics of the environmental impact of oil shale in situ mining. The proposed evaluation model is described as $P_A = 0.464 \times (0.197P_{C1} + 0.198P_{C2} + 0.253P_{C3} + 0.352P_{C4}) + 0.289 \times (0.515P_{D1} + 0.285P_{D2} + 0.2P_{D3}) + 0.247 \times (0.385P_{E1} + 0.257P_{E2} + 0.35P_{E3})$;

(2) The following carbon-emission-related elements are introduced in the three-level indicator calculation model: biological carbon fixation, carbon emissions, carbon reduction performance, and carbon emission monitoring and control, with assessment scores of 70, 61, 80, and 75, respectively. Among them, the assessment score of carbon emissions is the lowest, and carbon emissions in all development links require control;

(3) The Fuyu oil shale mining area was assessed through the proposed comprehensive evaluation model, and the evaluation results of the mining area under two different heating methods, combustion heating and electric heating, were compared. The calculation results under the combustion heating method show that the highest hydrogeological evaluation score is determined as 88, and the lowest ecological sensitivity evaluation score is 48.1. The three-level indicator of ecological sensitivity requires improvement, and the final PA is 72.6. Similarly, the $P_A$ value obtained under the electric heating method is 70.5. The results indicate that the environmental impact of in situ mining under the combustion heating method in the region is smaller than that under the electric heating method.

**Supplementary Materials:** The following supporting information can be downloaded at: https://www.mdpi.com/article/10.3390/su16041363/s1, Table S1: Weights of the environmental capacity sub-index. Table S2: Grading of sub-index assessment values. Table S3: Sub-index weights of groundwater risk and prevention. Table S4: Hazard grade judgment of hazardous substances and process systems (P). Table S5: Division of environmental risk potential of construction projects. Table S6: Classification of assessment value of groundwater risk and prevention sub-indices. Table S7: Weights of cleaner production sub-indices. Table S8: Carbon emission factors corresponding to different energy sources. Table S9: Classification of the assessment value of the cleaner production

sub-index. Table S10: Pollution control sub-index weights. Table S11: Classification of assessment values of pollution control sub indicators. Table S12: Weights of process control sub indicators. Table S13: Grading of process control assessment values. Table S14: Differences in importance of secondary indicators in relative target layer. Table S15: Differences in importance of third-level indicators for relative environmental site selection. Table S16: Differences in the importance of third-level indicators of relative environmental risk. Table S17: differences in importance of third-level indicators for relative environmental governance.

**Author Contributions:** Conceptualization, H.Y. and B.L.; methodology, S.H.; validation, D.J. and Y.S.; formal analysis, D.J.; investigation, L.Q.; resources, S.H.; data curation, X.W. and B.L.; writing—original draft preparation, X.W.; writing—review and editing, X.W. and B.L.; visualization, W.X.; supervision, H.Y.; project administration, Y.S.; funding acquisition, S.H. All authors have read and agreed to the published version of the manuscript.

**Funding:** This research was partly funded by the national key research and development project "Ecological environment and geological effect evaluation of in situ mining of oil shale" (2019YFA0705504) from the Ministry of Science and Technology, PRC.

**Institutional Review Board Statement:** Not applicable.

**Informed Consent Statement:** Not applicable.

**Data Availability Statement:** Not applicable.

**Conflicts of Interest:** Authors Shaolin He, Yang Song and Wei Xu were employed by the company Beijing Zhonglu Consulting Co., Ltd. The remaining authors declare that the research was conducted in the absence of any commercial or financial relationships that could be construed as a potential conflict of interest.

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
