# Peer review of "An Analytic Hierarchy Process Method to Evaluate Environmental Impacts of In Situ Oil Shale Mining"

_sustainability, doi:10.3390/su16041363_

Round 1

Reviewer 1 Report

Comments and Suggestions for Authors

In item 1 – Introduction, there should be a more dense explanation of what shale oil exploration is, that is, from line 29 to line 109.

Water quality and pyrolysis are discussed.  Soon after, the discussion of the literature review begins, commenting on what has been done in the works of other authors, as well as the methods used.  Then there's the discussion about carbon emissions and references to other industrial processes.

The work would be better presented if in introduction had a discussion of the shale extraction process and then the model had been presented.  I think that this introduction, made in a direct way, is absent from the text.

In the other parts of the work, the presentation is made in a coherent way.

Reviewer 2 Report

Comments and Suggestions for Authors

This research examines an analytical approach derived from prior literature and employs the Delphi technique. A set of three secondary indicators and ten tertiary indicators were chosen to construct an evaluative index system. The paper is worth publishing however, it requires some improvements before publication:

1-     Literature review must be improved by introducing the recent achievements in the ecological environment assessment index system of oil shale in-situ mining.

2-     The available gaps in the literature have not been indicated clearly.

3-     In the section 1. Introduction the authors discuss the possibility to derive the carbon emission reduction potential in production sites. Notably, novel cost-effective intelligent approaches have emerged to tackle issues linked to the monitoring of different parameters (https://doi.org/10.1016/j.csite.2023.102778) and (https://doi.org/10.1016/j.heliyon.2023.e17282).It would be interesting that authors have a review of these techniques in the first chapter of the paper.

4-     Figures 1 and 2, contains too much information. It would be better to present only the principal boxes.

5-     In the section 5.4.1. it might be clearer to present the sub-indicators in a figure.

6-     Removing un-necessary sub-sections of the Chapter 6.

7-     Limitation of the study and possible future research must be indicated in the end of the paper.

8-     The conclusion must be improved.

Reviewer 3 Report

Comments and Suggestions for Authors

The method is very limited and the idea of this paper has no contribution to recent scholarship, moreover, there is no discussion and results analysis, so this is not a formal academic paper.

Reviewer 4 Report

Comments and Suggestions for Authors

In their manuscript "Research on AHP based comprehensive assessment model for 2 the eco-environment of the in-situ low-carbon exploitation of 3 oil shale", X. Wang and colleagues present a multi-criteria framework to assess the environmental impacts of oil shale exploitation.

In the last decades, the analytical hierarchy process has been extensively applied to a variety of environmental assessment problems. Its application to the case study presented by the authors seems appropriate to me. However, the proposed assessment framework fails, in my opinion, to go deep into the problem. There is a lack of a robust discussion on how the criteria were selected. What are the main environmental implications of oil shale exploitation? How do the selected indicators take those implications into account? Most of the indices used for the evaluation of the third-level criteria are themselves multi-criteria indices and seem to have been developed by the authors, but the justification for their structure is barely argued. Furthermore, multi-criteria evaluations make sense for comparing alternatives across a broad spectrum of dimensions rather than for providing an absolute evaluation of a single alternative. In the present case, however, the authors derive a single figure that should characterise the environmental conditions of the specific case study (the Fuyu oil shale mining area). The value obtained (72.6) is claimed to be moderate, but this evaluation is in my opinion of little significance without references. What value would another mining area obtain? What would be the value of the area studied if mining were conducted in another way? What if the area was not mined instead? I am sorry but in my opinion the article is not publishable as it stands now. The authors should better justify their choices, discuss the structure of their indicator better and more critically, and provide a comparative evaluation of more alternatives.

Here are a few comments that I hope will help improve the text for this or another journal.

Title:

I suggest rewording the title more clearly, perhaps something like “An AHP approach to assessing environmental impacts of oil shale exploitation”

Abstract:

L. 13:     “ecological environment”: As an ecologist, I find the authors' systematic use of the term ecology misleading. In the proposed framework, ecological aspects, understood as relating to impacts on the way organisms interact with each other and their environment, are indeed marginal (see my comments below). Therefore, I suggest replacing occurrences of “ecological environment” simply with “environment”.

L. 15:     The authors mention an “evaluation index system”, a “three-level indicator evaluation model”, a “comprehensive evaluation index”, a “comprehensive evaluation model” as if they were different metrics but, if I understand correctly, there is only one index obtained through the weighted aggregation of seven environmental indices. Please be consistent in the naming to avoid confusing the reader.

L. 22:     It is not clear what the “target layer” is, please be clearer

Introduction

L. 33:     It am somewhat surprised to read that there is a “current global energy shortage” in a journal called Sustainability, I would rather say that there is an “growing demand for energy” (despite the evidence that we should find a way to stop it, but this is probably a personal position).

L. 37:     Again, talking about the “environmental friendliness” of mining sounds very strange, at least in an absolute sense (one can say that one mining technology is more environmental-friendly than another, but the activity itself does not improve the quality of the environment so I would never say it is environmentally friendly).

L. 44:     “in order” → “In order”

L. 45:     “Ultrapure” → “ultrapure”

L. 61:     The “comprehensive index evaluation method” is not, to my knowledge, a commonly used environmental assessment method. The cited reference (Zhang et al. 2013) is a conference abstract providing no information on the method. A quick search on the topic gave no further result. In addition, the PSR model (or its extension, the DPSIR model) is an approach used to conceptualize assessment problem rather than an assessment method itself. AHP and Delphi are decision support techniques that are actually used in the context of environmental impact assessment (and many other decision-making problems), while the PCA is a statistical technique for reducing the dimensionality of a dataset, although it can be used to support criteria weight assignment in multi-criteria decision making. I think this background part should be made more robust.

L. 83:     I suggest simplifying the sentence as “analysing the environmental impacts of oil shale in-situ mining”

L. 85:     I do not understand what the authors means exactly with “layer”

L. 87:     This paragraph seems to be related quite loosely with the rest of the introduction

L. 89:     “certain substances” is too vague a term for a scientific paper

Construction methods and steps of the oil shale index system

L. 111:   “A literature analysis method” → “A literature analysis”

L. 126:   “oil and gas is then extracted” → “oil and gas are then extracted”

L. 149:  

Calculation of index weights using the analytical hierarchy process

L. 165:   The pairwise comparison matrices are said to have been built via expert judgement, but the elicitation process is not described and the detailed results of expert involvement are not reported. Furthermore, it is not clear whether the construction of the judgement matrices was carried out before or after the definition of the indices. That is, whether the weights were generically assigned to the criteria or whether the experts were adequately informed about the structure of the indicators used to assess the different aspects considered. In addition, the references cited in the paragraph do not seem to be very relevant.

Construction of typical indices and comprehensive evaluation model

L. 193:   It is not clear what the “limitations of the more statutory index of AHP” are and how they were overcome

L. 199:   What is the “standard layer”?

L. 204:   The environmental capacity index uses the maximum amount of pollutant that meets the environmental standard as a reference point. However, I do not understand from the supplementary information how the assessment is made. Where does the impact of the activity in question appear in the formulas? The environmental standard and background pollutant concentrations are mentioned in the proposed equations, but where do the contributions of the specific activity appear? Perhaps I have not understood correctly, but I think the index needs to be explained better.

L. 289:   The equations in this paragraph are barely informative. Eqs. 3-6 and 7-11 could be reduced to a couple of generic equations as they all share the same structure

L. 323:   “construction of sewage” → “production of sewage”

L. 452:   The proposed assessment method should be used, in my opinion, to compare alternatives, not to say whether a single alternative is feasible or not.

L. 479:   Ecological sensitivity is the only index taking into account the ecological implications of the mining activity. However, the reference cited to support the choice of the index is not listed in the references section, so there is no way to assess whether the index is appropriate. Furthermore, the way in which the proposed index is applied to the specific case study seems rather approximate to me.

L. 484:   “the normalization coefficient” → “a normalization coefficient”

L. 502:   What is the “birth state sensitivity”?

Conclusion

L. 668:   Why should an index value of 72.6 be moderate? And why does this lead the authors to conclude that the evaluation model is appropriate?  

Comments on the Quality of English Language

Sometimes the text seems somewhat roughly translated from Chinese, I suggest a revision by a native English speaker

Round 2

Reviewer 2 Report

Comments and Suggestions for Authors

The authors have implemented all suggested comments, significantly enhancing the quality of the paper. As a result, the paper is now ready for publication.

Reviewer 3 Report

Comments and Suggestions for Authors

The author said: "Moreover, there are few reports on incorporating carbon emissions into environmental evaluation indicators", actually , there are many studies conducted to reveal environmental impact involving carbon emissions. The AHP method is a very old method and its contribution is very limited.

Reviewer 4 Report

Comments and Suggestions for Authors

I recognise that the authors made an effort to respond to my comments and to improve the quality of the manuscript. I remain puzzled by the final statement that, given the overall score assigned to the analysed project, this turns out to have an acceptable environmental impact. As I said in my previous review, the outputs of multi-criteria decision methods are of relative rather than absolute value, i.e. they can be used to compare options rather than to assess the overall sustainability of an option. What is the scientific basis for stating that a score between 75 and 90 corresponds to a minor impact? Score ranges are conventional and, unless they are based on specific environmental thresholds, saying that a given score is acceptable or not makes little sense.

Comments on the Quality of English Language

The quality of the English language can be further improved

Round 3

Reviewer 3 Report

Comments and Suggestions for Authors

The author had revised according to the review suggestions.